# Evaluating the Predictive Potential of Patient-Specific Biomechanical Models in Class III Protraction Therapy

**DOI:** 10.3390/bioengineering12111173

**Published:** 2025-10-28

**Authors:** Joeri Meyns, Wout Vertenten, Sohaib Shujaat, Sofie Van Cauter, Constantinus Politis, Jos Vander Sloten, Reinhilde Jacobs

**Affiliations:** 1Department of Oral and Maxillofacial Surgery, General Hospital St-Jan, 3600 Genk, Belgium; 2OMFS-IMPATH Research Group, Department of Imaging and Pathology, Faculty of Medicine, KU Leuven, 3000 Leuven, Belgium; sohaib.shujaat941@gmail.com (S.S.);; 3Department of Medicine and Life Sciences, Hasselt University, 3500 Hasselt, Belgium; 4Biomechanics Section, Department of Mechanical Engineering, KU Leuven, 3001 Leuven, Belgium; 5King Abdullah International Medical Research Center, College of Dentistry, King Saud Bin Abdulaziz University for Health Sciences, Ministry of National Guard Health Affairs, Riyadh 11481, Saudi Arabia; 6Department of Medical Imaging, General Hospital St-Jan, 3600 Genk, Belgium; 7Department of Dental Medicine, Karolinska Institute, 171 77 Stockholm, Sweden

**Keywords:** Class III malocclusion, finite element analysis, biomechanical modeling, maxillary protraction, treatment prediction

## Abstract

Predicting treatment outcomes in Class III protraction therapy remains challenging. Although finite element analysis (FEA) helps in the study of biomechanics and planning of orthodontic treatment, its use in Class III protraction has mainly been in evaluating appliance designs rather than patient-specific anatomy. The predictive accuracy of FEA has not been validated in Class III protration therapy. In this study, ten patients (5 female, 5 male, aged 7–11 years) with Class III malocclusion received either facemask or mentoplate treatment. CT scans from four patients were used to construct simplified finite element models, and predictions were compared with one-year treatment outcomes from six additional patients. While stress patterns differed between treatments, patient-specific geometrical factors had a more significant impact on deformation than treatment type. FEM-predicted maxillary changes (mean: 0.352 ± 0.12 mm) were approximately one-tenth of actual changes (mean: 1.612 ± 0.64 mm), with no significant correlation. Current FEM approaches, though useful for understanding force distribution, cannot reliably predict clinical outcomes in growing Class III patients. The findings suggest that successful prediction models must incorporate biological and growth factors beyond pure biomechanics. Accurate prediction of treatment outcomes requires comprehensive models that integrate multiple biological and developmental factors.

## 1. Introduction

Growing skeletal Class III malocclusion presents a complex challenge in orthodontics. The main difficulty lies in predicting growth potential of the maxilla and mandible. Early intervention decisions are complicated by uncertain skeletal growth patterns. Current debates focus on which patients are best suited for early Class III treatment [1,2,3,4].

Facemask (FM) therapy combined with rapid palatal expansion (RPE) is a common treatment approach [5]. The effectiveness depends on how forces transfer to the jaw. Traditional tooth-borne (TB) applications produce limited skeletal effects. Skeletal anchorage (SA) devices offer an alternative. These include mentoplate (MP) and palatal screws, such as Hybrid Hyrax [6]. SA devices may provide better skeletal results and vertical control, especially in high-angle patients [7,8,9,10].

Despite increasing use of SA devices, several questions remain unanswered. The optimal treatment age is unclear. Force levels are not standardized. Evidence comparing outcomes between SA devices and conventional methods is limited [11,12,13,14].

Initial treatment success doesn’t guarantee long-term stability. Some patients show deterioration in occlusion and facial aesthetics as they mature [4]. The key challenge is patient selection. Clinicians need to identify which patients will maintain positive outcomes from early intervention. They also need to determine which patients should wait for skeletal maturity and possible orthognathic surgery.

Predictive models exist using 2D cephalometric and CBCT measurements. However, most are based on retrospective studies without proper validation [15,16,17]. New approaches combine these measurements with initial treatment response [18]. The predictive accuracy of these methods remains uncertain [15].

Research on maxillary protraction has evolved significantly. Early studies used basic models like wax, elastic, and dry skulls. Modern studies employ sophisticated 3D imaging and finite element (FE) modeling. Early research established fundamental concepts about centers of resistance and rotation [19,20,21,22,23,24,25]. Current FE analysis has enhanced our understanding of force distribution, particularly with SA techniques [26,27,28,29].

Despite the potential of finite element (FE) studies in analyzing skeletal anchorage (SA) techniques, several critical limitations persist in current research. Existing studies present contradictory findings regarding skeletal effects and vertical control [27,28], with many analyses overlooking crucial aspects such as mandibular deformation [26,27,28,30]. Research has predominantly concentrated on appliance design optimization while neglecting patient-specific anatomical variations. Furthermore, while studies frequently describe deformation patterns and hypothesize about their relationship to treatment outcomes, a significant knowledge gap remains: No studies have validated these models against actual clinical outcomes in Class III protraction therapy. Treatment success depends on both biomechanical forces and complex biological responses. Current modeling approaches may not capture all these factors. Understanding whether FEM can reliably predict treatment outcomes is crucial. This knowledge would either validate current biomechanical approaches or reveal the need for more sophisticated predictive tools.

This study presents the first direct comparison between finite element model predictions and actual clinical outcomes in Class III protraction therapy. We tested two key hypotheses using image data from a previous randomized controlled trial. First, we examined whether simplified FEM could accurately predict stress distribution and deformation patterns corresponding to actual treatment effects. Second, we investigated whether different treatment techniques (FM vs. MP) would produce distinct patterns of skeletal deformation. Our findings challenge current assumptions about using biomechanical modeling in orthodontic treatment planning.

## 2. Materials and Methods

### 2.1. Trial Registration and Ethical Approval

We obtained CT scans from patients enrolled in a previous randomized controlled trial. This trial evaluated both 2D [31] and 3D [32] outcomes. The RCT was registered at www.ClinicalTrials.gov (ID: NCT02711111). The Ethics Committee at Ziekenhuis Oost Limburg, Belgium granted approval (EudraCT B371201629565) on 13 December 2016.

### 2.2. Subjects

Our study analyzed CT scans from ten Caucasian patients with Class III skeletal malocclusion. The group included five females and five males, aged 7–11 years. All patients participated in a randomized trial comparing Facemask (FM) (Figure 1) and Mentoplate (MP) (Figure 2) treatments [31,32]. Each patient received a Hybrid Hyrax apparatus (HH) assembled with two mini-screws in the anterior palate and an expansion screw attached to first molar bands (Figure 1 and Figure 2). We used the Alt-RAMEC protocol for expansion [33], wherein the HH was activated by the patient’s parents twice daily (0.25 mm per turn, two turns in the morning and two turns at night) for one week, followed by deactivation twice daily (two turns in the morning and two turns at night) for the next week. This cycle of alternating activation and deactivation was repeated three times. In the following week, the maxilla was adjusted to the suitable transverse dimension. The FM group received elastic forces of 360–400 g per side and wore the appliance 12–14 h daily. The MP group received 185 g force per side with continuous wear.

We randomly selected two patients from each treatment group to develop our simplified Finite Element Model (FEM). We then tested the model on six additional patients, three from each treatment group, to evaluate its predictive abilities.

Our methodology followed three main phases. First, we created patient-specific 3D models from pre-treatment CT scans. Next, we performed finite element analysis to simulate both treatments. Finally, we compared predicted outcomes with actual one-year clinical results. This approach allowed direct validation of FEM predictions against real treatment outcomes.

### 2.3. Creating a Patient Specific 3D FE Model

We obtained CT scans at treatment (T0) start using a Siemens Somatom Force scanner (Siemens^®^, Erlangen, Germany). The scanning protocol used a slice thickness of 0.6 mm, with 50 mA current and 150 kV voltage. Each scan took 2.04 s to complete.

We imported the DICOM files into Mimics^®^ software (Materialise^®^, Leuven, Belgium) for segmentation. To improve efficiency, we excluded several regions not critical to the analysis. These included the posterior midface, skull, teeth, and superior portion of the alveolar process (Figure 3). The teeth were also removed since forces were transferred through a bone-anchored device (Hybrid Hyrax), which has been proven to produce good skeletal anchorage with minimal dento-alveolar effect [10,27]. The periodontal ligament was not simulated, based on a previous study suggesting that modeling the periodontal ligament in finite element analyses of skulls can be ignored if the values of stress and strain in the alveolar region are not required [34]. Finally, the superior portion of the alveolar process was excluded because significant remodeling occurs in this region during tooth eruption, which could interfere with accurate comparisons to the one-year follow-up scan.

We transferred the 3D model to 3-Matic software (Materialise^®^, Leuven, Belgium) for meshing. A surface mesh was generated using specific edge lengths tailored to the anatomical features of the structures: 1 mm for the maxilla and 2 mm for the mandible. These sizes were chosen based on previous studies [27] and to balance accuracy and computational efficiency, as the maxilla’s finer bone structures required higher resolution, while the mandible’s thicker and less complex geometry allowed for a coarser mesh without compromising accuracy. Different triangle edge lengths were used to maintain anatomical accuracy. The meshing process also involved ensuring that the surface mesh was free from elements with poor computational properties, such as sharp internal angles. Following this, a uniform volume mesh type was applied to both jaws for consistency during finite element analysis (FEA). The final 3D meshed FE models, depicted in Figure 3C, represent the maxillary and mandibular structures in detail, for further processing.

### 2.4. Finite Element Analysis (FEA)

We conducted our analysis using Abaqus^®^ software (version 2020) (Dassault Systèmes^®^, Vélizy-Villacoublay, France). The model incorporated calculated force magnitudes and orientations along with appropriate boundary conditions. For this simplified FEM, the maxilla and mandible were modeled as uniform bone structures with isotropic and elastic material properties to ensure computational efficiency while maintaining physiological relevance. The assigned material properties were based on previous studies [26,27,28,29]. The maxilla was assigned a Young’s modulus of 700 MPa and a Poisson’s ratio of 0.3, representing its primarily trabecular bone composition, which is softer and more porous. In contrast, the mandible was assigned a Young’s modulus of 1000 MPa and a Poisson’s ratio of 0.3, reflecting its denser, cortical bone structure, which contributes to its increased stiffness.

### 2.5. Boundary Conditions

We applied boundary conditions for the upper jaw at the infra-orbital rim (Figure 4). This region remains stable during growth and treatment, making it an ideal reference point [35]. We fixed displacements in all three spatial directions at selected nodes while allowing rotation. The lower jaw presented more complex challenges for boundary conditions. Many regions undergo remodeling during growth. While the anterior chin and internal symphysis have been considered stable and reliable for voxel-based superimposition in growing patients [36], applying boundary conditions at these locations can overly constrain the model. Their high Young’s modulus combined with boundary constraints prevented realistic deformation under treatment forces. The condylar region also proved unsuitable due to significant remodeling during growth and treatment [37]. We performed a three-dimensional volumetric comparison analysis on four patients from our RCT. This analysis revealed the angle of the mandible as a relatively stable area. We therefore fixed three nodes at each mandibular angle to achieve optimal model stability while allowing deformation. This approach balances the mandible’s high stiffness (Young’s Modulus) with realistic movement, as more constraints would prevent deformation and fewer would create instability (Figure 4).

### 2.6. Magnitude and Vector of the Forces

The MP therapy used continuous low-force elastics generating approximately 200 g of force per side [6,9,38]. We calculated precise force vector directions using 3-Matic software by analyzing the line between the MP’s mandibular attachment point and molar anchor points. This patient-specific calculation ensures accurate modeling of force orientation in FE simulations. FM therapy required higher force elastics generating approximately 400 g per side, worn 12–14 h daily. We positioned the force vector at 25–30° to the occlusal plane [1,39,40]. Both treatments incorporated Alt-RAMEC using a HH expander with dual palatal mini-screws. This device generated approximately 2500 g of force during palatal expansion [41,42,43].

### 2.7. Force Simulation

The forces acting on the upper and lower jaws were applied to the FE model through node sets to ensure accurate simulation of treatment forces (Figure 4). For the upper jaw, we selected two primary node sets based on the HH device design. The first set resided in the first molars’ apical region. The second set included nodes near the anterior palatal screws. This configuration allowed for realistic simulation of both FM and MP therapy forces in the upper jaw.

In the lower jaw, the method of force application differed between FM and MP therapies. For FM therapy, 43 nodes were selected within the chin-cup contact area (Figure 4a). These nodes were distributed to cover the interaction zone where the FM applies pressure to the mandible, ensuring that the force transmission was realistically simulated. For MP therapy, each screw of the MP was represented by six nodes—three on the external surface and three internally within the mandible (Figure 4b). This arrangement was designed to accurately model the dispersive effect of forces as they propagate through the bone tissue surrounding the screws. The models incorporated realistic forces, with MP therapy requiring twice the elastic wear time of FM therapy. The number of deformation cycles was adjusted accordingly for each treatment to match actual clinical usage patterns. To capture these variations, the FM models were subjected to 5 deformation cycles, while the MP models underwent 10 cycles. Abaqus software enabled the iterative reapplication of forces to the already deformed geometry of the models, further enhancing the patient-specific accuracy of the FEA. This approach ensured that the simulation realistically represented the cumulative deformation and stress patterns corresponding to each treatment protocol.

The differences between the deformed and baseline models were analyzed using part comparison analysis (PCA). This technique produces color-coded maps that visualized the extent of deformation across various regions of the jaw structures. By incorporating these methods, the simulation captured the biomechanical behavior of craniofacial structures under treatment, providing insights into force distribution and the resulting deformations with high precision. Appendix A provides more details on node sets and forces applied on these nodes.

### 2.8. Actual Treatment Effect

The baseline (T0) and one-year follow-up (T1) CT datasets were imported into Amira^®^ software (version 2019.1, Thermo Fischer Scientific^®^, Merignac, France) in DICOM format to analyze skeletal changes over the treatment period. Using volume rendering, the datasets were visualized in 3D to enable precise identification of structural differences (Figure 5). A rigid voxel-based registration was performed with mutual information as the alignment metric, ensuring accurate superimposition of the datasets [44,45]. The T1 dataset was registered to T0 using stable anatomical landmarks: the inferior orbital rim for the maxilla and both the external oblique ridge and mandibular angle for the mandible. The inferior orbital rim’s stability is well-documented in the literature [35], while our previous PCA analysis confirmed the stability of the mandibular landmarks during growth. We then imported the registered datasets into Mimics^®^ software (version 20.0, Materialise^®^, Leuven, Belgium) for segmentation. A semi-automatic thresholding approach, refined with manual adjustments, was employed to construct high-resolution 3D volumetric surface models of the maxilla and mandible. These models were then exported in STL format (Figure 5), preserving the geometric fidelity required for further analysis.

The STL models were subsequently transferred to 3-Matic^®^ software (version 14.0, Materialise^®^, Leuven, Belgium) for part comparison analysis. This process calculated the mean differences between the T0 and T1 datasets, quantifying treatment-induced changes in the skeletal structures. The results were visualized as a color-coded map (Figure 5), with different colors representing the magnitude and distribution of deformation.

### 2.9. Comparing Modeled Deformation with Actual Deformation

We evaluated FEM accuracy by comparing predicted (PCAmD) and actual (PCAaD) deformations using visual and quantitative methods. Color-coded maps illustrated deformation patterns. Initial analysis of four FEM cases showed promising maxillary results, but mandibular comparisons proved impossible due to model limitations.

We expanded the study with six additional patients. We focused on maxillary deformation using five anatomical landmarks per patient for qualitative analysis (Figure 6, Table 1). Our analysis revealed that predicted deformation values were notably smaller than actual measurements. The correlation between PCAmD and PCAaD at these landmarks provided insight into model accuracy.

## 3. Results

### 3.1. Stress-Distribution

#### 3.1.1. Maxillary Stress Distribution (Figure 7)

Both the FM and MP treatment groups exhibited high local stress just below the zygomatic bone, specifically in the area apical to the first molar, where the applied forces are transferred to the bone. Despite the FM group experiencing elastic forces that were twice as strong and originating from the same anchor point as the MP group, the resulting stress patterns were similar between the two treatments. However, the magnitude of the stresses varied among patients, corresponding to the thickness of the cortical bone. Thinner bone in this region correlated with greater stress distribution.

**Figure 7 bioengineering-12-01173-f007:**
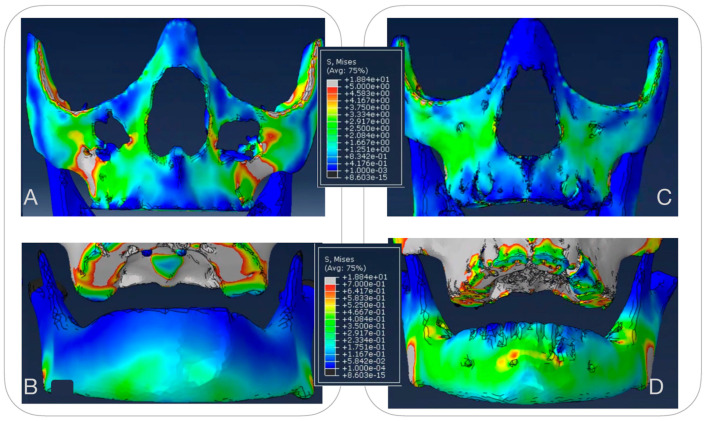
Von Mises stress distribution in the jaws. Panels (**A**,**C**) (upper jaw): Stress ranges from 0.001 MPa (dark blue, minimal stress) to 5 MPa (bright red, maximal stress). Panels (**B**,**D**) (lower jaw): Stress ranges from 0.0001 MPa (dark blue, minimal stress) to 0.7 MPa (bright red, maximal stress). The color scale indicates the relative magnitude of stress, with cooler colors representing lower values and warmer colors representing higher values.

Maxillary stress distribution varied from 0.001 Mpa to 5 Mpa, with all MP models exhibiting a small region of very high stress below the zygomatic bone, precisely where the elastic forces originated. This difference in stress distribution was treatment-specific rather than patient-specific. It is likely attributable to the slightly more medial and less downward vector of the forces in the MP group (averaging 25 degrees) compared to the FM group (averaging 30 degrees downward). These subtle differences in force direction highlight the impact of treatment mechanics on stress distribution within the craniofacial structures.

#### 3.1.2. Mandibular Stress Distribution (Figure 7)

The scales in Figure 7 indicate that stresses in the mandible are ten times smaller than those in the maxilla, ranging from 0.0001 Mpa to 0.7 Mpa. This difference is attributed to the mandible’s thicker cortical bone, which has a higher Young’s modulus. The primary stress regions in the mandible are located in the symphyseal area. Additionally, the stress distribution patterns in the mandible differ between the two treatment options. In the FM group, stress in the symphyseal region is more evenly distributed along the entire mandible. In contrast, the MP group exhibits higher and more localized stresses concentrated around the area where the MP is attached to the bone.

Interestingly, one patient-specific model showed unilaterally higher stress distribution in both the upper and lower jaws. This finding could only be attributed to individual patient-specific characteristics.

### 3.2. Modelled Deformation (PCAmD)

Looking at the overall deformation, maxillary deformation patterns revealed a patient-specific response (Figure 8), with less deformation observed in patients with thicker cortical bone, regardless of the treatment type. Although stress patterns differed between the FM and MP groups, these differences did not influence maxillary deformation. In contrast, mandibular deformation exhibited clear treatment-specific patterns, with the MP group showing greater deformation than the FM group. The model deformation was analyzed both overall and along three coordinate axes: transverse (x), sagittal (y), and vertical (z). On the x-axis, positive values indicate leftward movement, negative values rightward movement. On the y-axis, positive values represent backward movement, negative values forward movement. On the z-axis, positive values show upward movement, negative values downward movement. The results display displacement patterns using a color-coded system, with red indicating maximum displacement and blue showing minimum displacement. Figure 8 illustrate the displacement patterns for all four patients across three axes. In the x-axis, maximum deformation occurs in the posterior maxillary region apical of the first and second molars. These deformations ranged from 0.27 mm to the left to 2.74 mm to the right and were patient-specific. The midface shows minimal deformation, and the mandible remains largely stable except for minor condylar changes. Treatment-specific effects are evident in both jaws in the y-axis: Facemask patients show greater upper jaw deformation and mentoplate patients display greater lower jaw deformation. Mentoplate patients showed more anterior mandibular movement than facemask patients. This movement likely results from counterclockwise mandibular rotation around the mandibular angle, explaining backward condylar displacement. The z-axis analysis confirms these treatment-specific vertical differences in the mandible. Interestingly, deformation in the upper jaw also showed counterclockwise rotation due inferior displacement of the posterior part of the palate, which was not treatment specific (downward displacement ranged from 0.57 to 1.2 mm).

### 3.3. Comparing Modeled Deformation (PCAmD) with Actual Deformation (PCAaD)

The modeled values (PCAmD) were found to be ten times smaller than the actual one-year deformation analysis (PCAaD) (Figure 9 and Figure 10). A patient-by-patient comparison revealed discrepancies and similarities. Cases with the highest predicted deformation occasionally exhibited lower actual changes, while cases with similar predicted patterns showed widely varying clinical outcomes. Notably, no consistent relationship could be identified between the model predictions and treatment success in the upper jaw (Figure 10). Further quantitative analysis of six additional upper jaws, based on five anatomical landmarks per patient, confirmed the absence of correlation between PCAmD and PCAaD. PCAmD values ranged from 0.110 to 0.509 mm (mean: 0.352 ± 0.12 mm), whereas PCAaD values ranged from 0.429 to 3.116 mm (mean: 1.612 ± 0.64 mm) (Table 1). Statistical testing revealed a significant negative correlation between PCAmD and PCAaD (Pearson’s r = −0.517, *p* = 0.003; Spearman’s ρ = −0.406, *p* = 0.026), indicating that higher modeled values were paradoxically associated with lower observed deformations.

Matching mandibular deformation with actual treatment effect proved more complex than analyzing those of the maxilla due to the lack of a fixed point of reference on the mandible. Additionally, the observed mandibular changes were consistently smaller than the finite element analysis (FEA) predictions, suggesting that our reference of boundary condition obscured critical details. As a result, comparing modeled deformation with actual deformation for the mandible was not feasible within the limitations of the current FE model (Figure 9).

## 4. Discussion

Early Class III treatment poses significant challenges, with initial positive outcomes often deteriorating over time despite clinicians’ best efforts [4]. As patients reach full maturity, treatment stability becomes a critical concern, potentially compromising both occlusion and facial aesthetics. Current research indicates that approximately 25–30% of patients who receive early Class III protraction therapy may ultimately require orthognathic surgery, highlighting the limited predictability of long-term treatment outcomes [46,47]. Identifying patients likely to benefit from interceptive treatment with minimal risk of relapse is critical. For those unlikely to respond well, delaying treatment until growth completion followed by orthognathic surgery may be a more effective strategy. Predictive factors for treatment success are often derived from 2D cephalometric analyses or linear/angular measurements from cone-beam computed tomography (CBCT) scans [15,16,17]. However, most of this research is retrospective and lacks proper validation on new cases, making their predictive accuracy uncertain [15]. Given the critical need for reliable prediction models in Class III protraction therapy and the current lack thereof, this study explored a three-dimensional approach to evaluate whether patient-specific finite element modeling could serve as a potential predictive tool. While FEM is commonly used to study tooth movement in orthodontics [48], few studies have validated their predictions against actual clinical outcomes [49,50]. Current FEM research on Class III protraction therapy typically uses idealized models rather than patient-specific data, and lacks clinical validation [26,29,51]. A lot of variability can be observed in FEM exploring the effect of Class III protraction therapy, with varying degree of detailing visualizing bone, sutures, periodontal ligament and teeth. Most assume isotropic homogeneous properties, despite evidence suggesting more complex characteristics, especially in the upper jaw [52]. This simplification may affect model accuracy at the individual patient level. Although some previous studies have attempted to model both jaws simultaneously [29], they typically lack detailed TMJ representation and some rely on adult dry skull specimens rather than age-appropriate samples [27]. Our simplified FEM deformation largely aligns with previous studies on maxillary protraction using skeletal anchorage devices [27,28,29,51]. Despite using different boundary conditions for the upper and lower jaw, we observed similar deformation patterns, specifically a counterclockwise (CCW) rotation of the craniomaxillary complex. The maxilla moves forward with downward displacement of the posterior palate, causing CCW rotation of the upper jaw [27,28,29]. Our mentoplate FEM shows CCW rotation of the lower jaw with anterior displacement of the mandibular symphysis, consistent with previous research using symphyseal plates [29]. However, the literature shows some contradictions. Some studies report less deformation with skeletal anchorage (SA) devices compared to tooth-borne (TB) solutions [28], while others show similar results [27]. Palatal anchorage devices may produce greater deformation [27]. Studies also differ on rotational effects, with some showing less CCW rotation in SA versus TB [28], while others indicate similar vertical control [27]. Given these inconsistencies, we should be cautious about drawing clinical conclusions from these studies. Additionally, no research has yet compared FEM predictions with actual treatment outcomes in Class III protraction therapy.

This study is the first to directly compare biomechanical model predictions with clinical outcomes in Class III protraction therapy. Our findings reveal no correlation between predicted and actual treatment effects, with clinical results approximately 10 times greater than model predictions. Figure 8, Figure 9 and Figure 10 illustrate these comparisons, with color-coded visualizations highlighting differences in deformation magnitudes and directions. The analysis revealed areas of alignment as well as discrepancies, shedding light on both the strengths and limitations of the FE modeling approach. This discrepancy suggests that current FEM approaches for Class III protraction therapy, while valuable for understanding force distribution, may be insufficient for clinical decision-making in growing patients. While the upper jaw deformation patterns aligned with previous studies, the model’s simplicity omitted crucial elements such as sutures, teeth, and periodontal ligaments. The lower jaw model proved particularly unreliable mainly due to oversimplified boundary conditions at the mandibular angle. This discrepancy highlights significant model limitations: 1. Oversimplified anatomy (omitting sutures, teeth, and periodontal ligaments). 2. Separate analysis of upper and lower jaws, ignoring their interconnected nature and the documented remodeling of the mandibular condyle and glenoid fossa during treatment [53]. 3. Assumption of homogeneous bone properties. 4. Exclusion of biological processes (growth, bone remodeling and post-treatment relapse). 5. Omission of muscular, masticatory, and soft tissue effects. These limitations suggest the need for more sophisticated models incorporating patient-specific data and biological processes. Another limitation of this study was the absence of mesh convergence testing, which is essential to validate the numerical stability and accuracy of finite element models. The selected mesh sizes were based on prior studies and anatomical considerations; however, future research should include systematic convergence testing to ensure that the results are independent of mesh density. Such testing will enhance the reliability of finite element models for clinical applications. While applying forces directly to nodes simplifies the modeling process, the authors recognize that selecting a surface, coupling it to a reference point, and applying the force to the reference point is an alternative approach that can more accurately distribute forces over a region. Future studies may explore this method to further refine loading conditions and improve the biomechanical representation of force applications in FEMs.

The study reveals that patient characteristics, particularly bone thickness, had a greater impact on deformation patterns than the type of treatment used. FM and MP treatments showed similar effects, indicating treatment choice does not significantly influence deformation patterns. Modeling the mandible accurately is difficult due to its complex structure and the temporomandibular joint (TMJ) mechanics. While the mandibular angle seems ideal for setting boundary conditions, applying constraints here can create artificial forces. To address this, developing a detailed TMJ model in Abaqus^®^ could better simulate mandibular movement. Although our simplified finite element model (FEM) showed deformation patterns consistent with previous studies on Class III protraction therapy, it could not accurately predict treatment outcomes. Ultimately, patient-specific factors, not treatment type, were most influential in deformation pattern.

Although this proof-of-concept study involved a small sample size, the consistent discrepancy between model predictions and clinical results underscores the reliability of the findings. Future research should expand to include a larger patient cohort and consider additional factors beyond anatomical properties.

## 5. Conclusions

Our systematic analysis highlighted fundamental limitations in current modeling approaches for Class III protraction therapy, particularly those relying solely on anatomical models. Although our simplified FEM had substantial limitations, deformation patterns aligned with previously reported Class III protraction therapy models. These findings underscore the inadequacy of purely biomechanical simulations in capturing the complexity of craniofacial treatment dynamics. Importantly, the analysis revealed that patient-specific characteristics, rather than treatment type, play a dominant role in determining deformation patterns. This suggests that successful predictive tools must integrate a broader range of factors beyond biomechanics alone.

## 6. Clinical Relevance

This study demonstrates that current finite element modeling, while useful for understanding force distribution, cannot reliably predict individual treatment outcomes in Class III malocclusion. Patient-specific characteristics, rather than appliance choice, appear to have greater influence on treatment success. Until more sophisticated predictive tools are developed, clinicians should view biomechanical simulations as Appendix A rather than definitive predictors when planning Class III treatment.

## Figures and Tables

**Figure 1 bioengineering-12-01173-f001:**
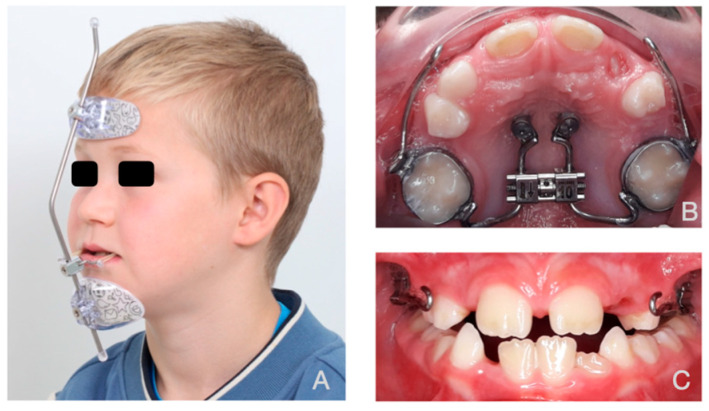
Facemask. (**A**): Extra-oral view. (**B**): Occlusal view (Hybrid Hyrax). (**C**): Intra-oral view.

**Figure 2 bioengineering-12-01173-f002:**
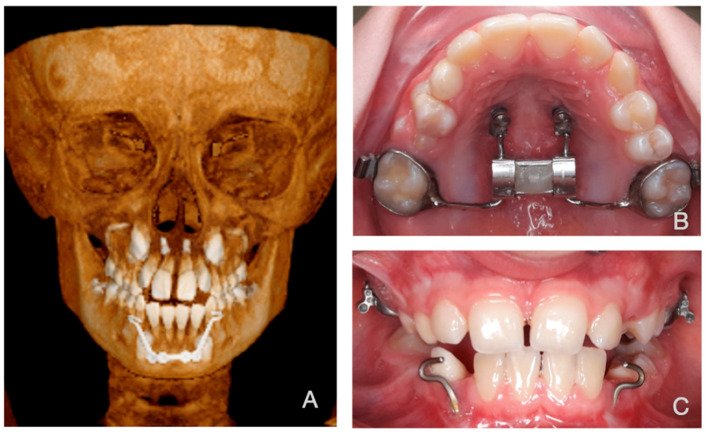
Mentoplate. (**A**): 3D volumetric frontal view (derived from CT data). (**B**): Occlusal view (Hybrid Hyrax). (**C**): Intra-oral view with the hooks of the mentoplate clearly visible.

**Figure 3 bioengineering-12-01173-f003:**
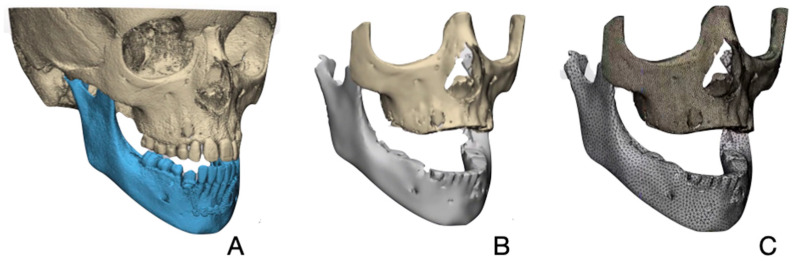
(**A**) Original DICOM dataset model. (**B**) Modified model with teeth, superior part of alveolar process, posterior midface, cranial base and skull region removed. (**C**) Finite element model showing detailed anatomical representation, with 2 mm mesh for mandible and 1 mm mesh for maxilla.

**Figure 4 bioengineering-12-01173-f004:**
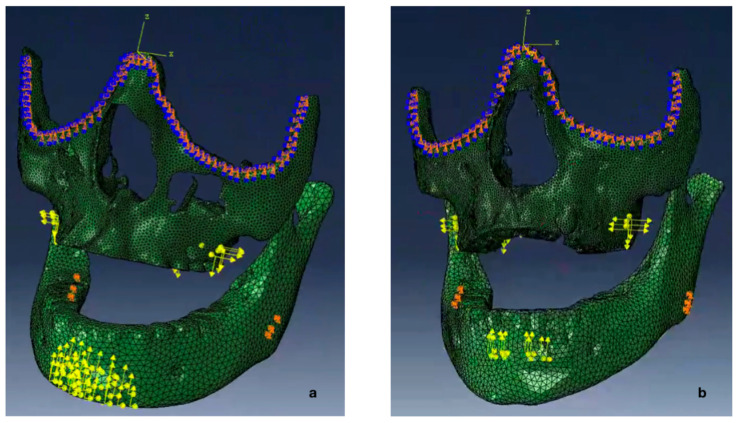
Two finite element models show maxillary protraction using: (**a**) Facemask and (**b**) Mentoplate. Yellow markers indicate force application points, while orange markers show boundary conditions at the infra-orbital rim (upper jaw) and mandibular angle (lower jaw).

**Figure 5 bioengineering-12-01173-f005:**
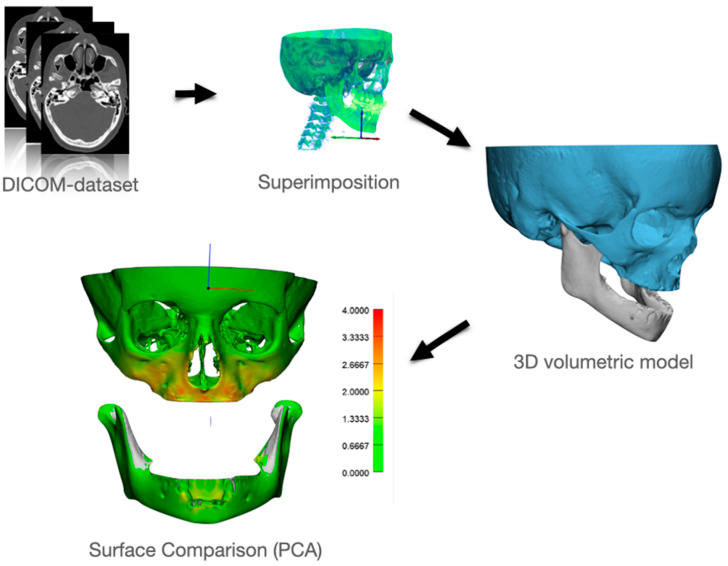
Workflow of Part Comparison Analysis (PCA): Both datasets are super-imposed (Amira^®^), Segmentation is performed in Mimics^®^ (semi-automatic) and exported in a 3D volumetric model (stl). Surface comparison (PCA) is done in 3-Matic^®^ and color-coded images are generated.

**Figure 6 bioengineering-12-01173-f006:**
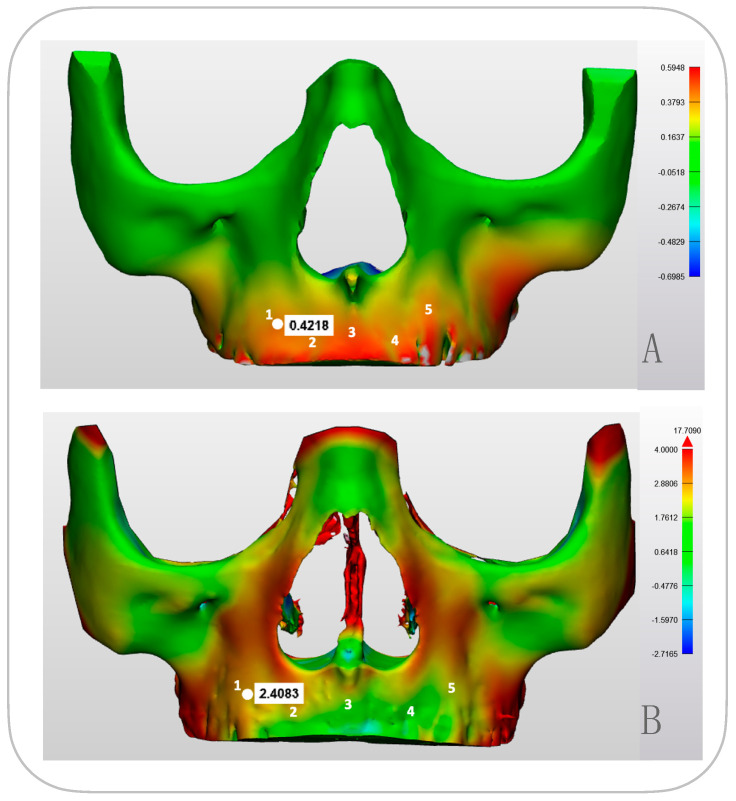
(**A**) illustrates the modeled deformation (PCAmD) through a color-coded map based on simulations. (**B**) illustrates the actual deformation (PCAaD) measured from an actual patient (FM-patient). Each color-coded map represents varying degrees of displacement, with the corresponding displacement values indicated by the scale (in mm) depicted next to each visualization. The maps clearly illustrate differences between simulated (predicted) and actual measured deformation. Five points are marked on each model for quantitative comparison between the simulation and the real-world results (numbers depicted in white 1–5).

**Figure 8 bioengineering-12-01173-f008:**
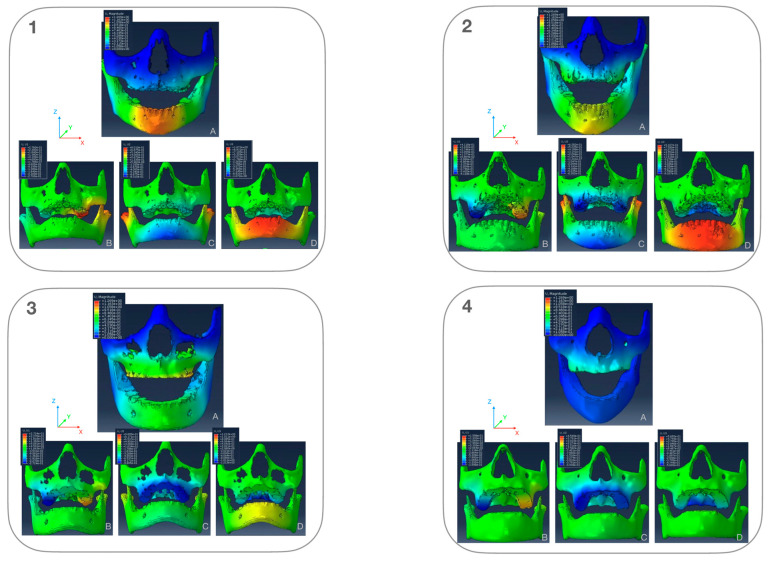
Color coded maps of deformation pattern in the FEM of the jaws in frontal view for four patients. Panel (**A**) shows the overall displacement pattern. Panels (**B**–**D**) show displacement along the three coordinate axes: (**B**). x-axis (lateral), (**C**). y-axis (sagittal), (**D**). z-axis (vertical). Patients 1 and 2 were treated with a mentoplate, while patients 3 and 4 were treated with a facemask. Each color-coded map represents varying degrees of displacement, with the corresponding displacement values indicated by the scale (in mm) depicted next to each visualization. scale (in mm) depicted next to each visualization.

**Figure 9 bioengineering-12-01173-f009:**
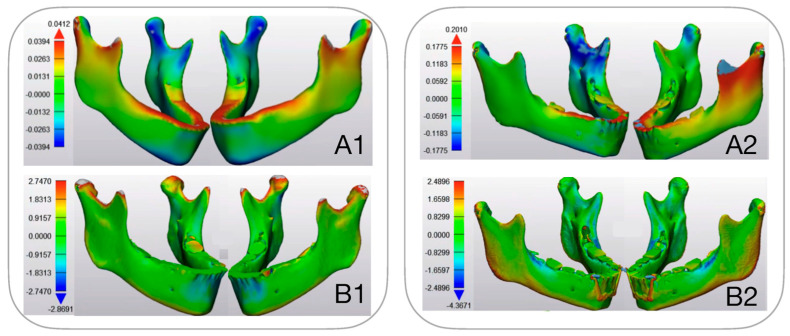
(**A**) Three-dimensional FEM showing displacement in color coded map of modelled deformation (PCAmD) and (**B**) color coded map of actual deformation (PCAaD). Patient 1 received facemask treatment and patient 2 received mentoplate treatment. Each color-coded map represents varying degrees of displacement, with the corresponding displacement values indicated by the scale (in mm) depicted next to each visualization.

**Figure 10 bioengineering-12-01173-f010:**
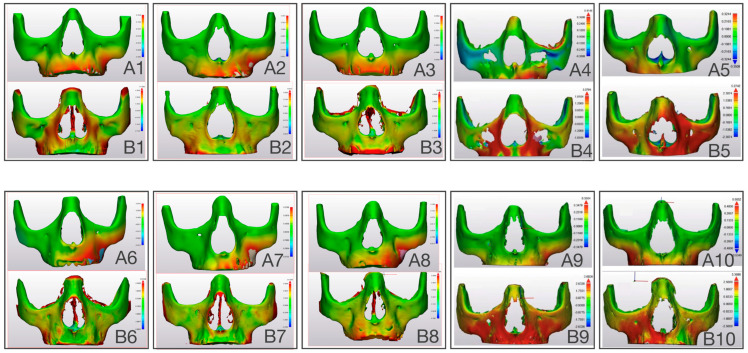
(**A**) Three-dimensional FEM showing displacement in color coded map of modelled deformation (PCAmD) and (**B**) color coded map of actual deformation (PCAaD). Patients 1–5 received facemask treatment and patients 6–10 received mentoplate treatment. Each color-coded map represents varying degrees of displacement, with the corresponding displacement values indicated by the scale (in mm) depicted next to each visualization.

**Table 1 bioengineering-12-01173-t001:** Measurements (mm) of changes at 5 anatomical points.

	Point 1		Point 2		Point 3		Point 4		Point 5	
	PCAmD	PCAaD	PCAmD	PCAaD	PCAmD	PCAaD	PCAmD	PCAaD	PCAmD	PCAaD
MP 1	0.22	1.75	0.33	1.50	0.35	1.57	0.51	1.03	0.49	1.90
MP 2	0.22	0.43	0.30	1.42	0.38	1.16	0.46	0.71	0.34	0.59
MP 3	0.11	3.12	0.15	2.42	0.15	2.73	0.16	2.08	0.18	2.84
FM 1	0.42	2.41	0.43	2.33	0.48	0.75	0.43	0.69	0.41	1.78
FM2	0.39	0.90	0.49	1.66	0.46	1.79	0.48	1.55	0.38	1.28
FM 3	0.30	2.21	0.35	1.85	0.43	1.10	0.34	1.48	0.43	1.37
mean	0.28	1.80	0.34	1.86	0.37	1.52	0.40	1.25	0.37	1.63
SD	0.12	1.00	0.12	0.42	0.12	0.70	0.13	0.54	0.11	0.75

PCAmD: modelled deformation; PCAaD: actual treatment effect after 1 year (PCAaD); FM: facemask; MP: mentoplate; SD: standard deviation.

## Data Availability

The original contributions presented in this study are included in the article and Appendix A. Further inquiries can be directed to the corresponding authors.

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
