# Peer review of "Evaluating the Predictive Potential of Patient-Specific Biomechanical Models in Class III Protraction Therapy"

_bioengineering, 2025, doi:10.3390/bioengineering12111173_

Round 1
Reviewer 1 Report
Comments and Suggestions for Authors
The authors are to be congratulated on their study, entitled "Evaluating the Predictive Potential of Patient-Specific Biomechanical Models in Class III Protraction Therapy." This study is considered to be of significant value as it is the first to compare the consistency of finite element models investigating the effectiveness of Class III treatment methods with clinical outcomes. The following two minor recommendations are hereby submitted for the consideration of the study's authors.
*Throughout the article, the capital letter "C" will be used to denote Class III.
*The information provided in lines 418-426 of the Discussion section is irrelevant in the present study as a tooth-borne appliance was not used.
Author Response
Comment 1: Throughout the article, the capital letter "C" will be used to denote Class III.
Response 1: Thank you for pointing this out. This was adjusted accordingly throughout the manuscript.
Comment 2: The information provided in lines 418-426 of the Discussion section is irrelevant in the present study as a tooth-borne appliance was not used.
Response 2: Thanks for the comment related to tooth-borne appliances, discussed in lines 562–568. Here we address the inconsistent findings reported in the literature, particularly with regard to comparisons between tooth-borne devices and skeletal anchorage. The sentence was deliberately retained in the manuscript to underscore these inconsistencies and to emphasize the need for caution when interpreting results.
Reviewer 2 Report
Comments and Suggestions for Authors
The paper Evaluating the Predictive Potential of Patient-Specific Biomechanical Models in Class III Protraction Therapy aims at using finite element analysis to predict the value of biomechanical modelling in orthodontic treatment planning.
The introduction and key words are appropiate.
However, the authors used only ten Caucasian patients in their paper. Is tis number of patient enough?
The comparison group also included a low number of cases.
The Alt-RAMEC used protocol should be described.
Why is this big gap between the date of the ethics committee protocol and the manuscript date: 13/12/2016?
The clinical relevance must be presented.
What is the originality of the study?
Figure 7 and 8 - Images of stress distribution must be explained.
How could future predictive models should incorporate a combination of biological markers, growth patterns, and patient-specific tissue responses to more accurately reflect the complex nature of craniofacial adaptation?
Author Response
Comment 1: However, the authors used only ten Caucasian patients in their paper. Is this number of patients enough? The comparison group also included a low number of cases.
Response 1: The image data employed in this analysis were originally obtained within the framework of a randomized controlled trial (RCT) with extended long-term follow-up (Meyns et al, Eur J Orthod 2025 Feb / Meyns et al. Prog Orthod. 2025 Apr). Computed tomography (CT) scans were acquired at different time points. As this investigation was designed as a proof‑of‑concept study, the primary objective was not the immediate validation of the finite element model (FEM). Such validation is inherently time‑consuming, and at the outset it was uncertain whether the model would adequately reflect the actual treatment effect. Consequently, the study was initiated with four patients and subsequently expanded to include an additional six, resulting in a total of ten participants. The authors acknowledge that this represents a limited sample size; however, this was a deliberate choice consistent with the exploratory nature of a proof‑of‑concept study. This limitation, and its implications, are explicitly addressed in the Limitations section (line 611 – 614):
Although this proof-of-concept study involved a small sample size, the consistent discrepancy between model predictions and clinical results underscores the reliability of the findings. Future research should expand to include a larger patient cohort and consider additional factors beyond anatomical properties
Comment 2: The Alt-RAMEC used protocol should be described.
Response 2: This was further explained in the revised manuscript under material and methods section (lines 101 – 106):
Each patient received a Hybrid Hyrax apparatus (HH) assembled with two mini-screws in the anterior palate and an expansion screw attached to first molar bands (Figure 1 and 2). We used the Alt-RAMEC protocol for expansion [33], wherein the HH was activated by the patient’s parents twice daily (0.25 mm per turn, two turns in the morning and two turns at night) for one week, followed by deactivation twice daily (two turns in the morning and two turns at night) for the next week. This cycle of alternating activation and deactivation was repeated three times. In the following week, the maxilla was adjusted to the suitable transverse dimension.
Comment 3: Why is this big gap between the date of the ethics committee protocol and the manuscript date: 13/12/2016?
Response 3: The image data utilized in this analysis were originally collected within the framework of a randomized controlled trial (RCT) with extended long-term follow-up (Meyns et al, Eur J Orthod 2025 Feb / Meyns et al. Prog Orthod. 2025 Apr). The finite element analysis (FEA) presented here was conducted as part of a doctoral dissertation associated with this RCT. As the dissertation was only recently completed, the present analyses were performed at this stage, despite the fact that the underlying image data had been available for a longer period of time.
Comment 4: The clinical relevance must be presented.
Response 4: The clinical relevance of the findings was further addressed in a Clinical Relevance section at the end of the manuscript (lines 631–637), where it was elaborated in greater detail:
This study demonstrates that current finite element modeling, while useful for understanding force distribution, cannot reliably predict individual treatment outcomes in Class III malocclusion. Patient-specific characteristics, rather than appliance choice, appear to have greater influence on treatment success. Until more sophisticated predictive tools are developed, clinicians should view biomechanical simulations as supplementary information rather than definitive predictors when planning Class III treatment.
Comment 5: What is the originality of the study?
Response 5: We thank the reviewer for this valuable comment and the opportunity to further clarify the originality of our work. In the revised manuscript, the Discussion section (lines 570–571) has been expanded with additional references to recent literature to highlight the novelty of this study as the first direct validation attempt in this field. It is indeed exceptional to have access to a prospectively collected three‑dimensional dataset of children during growth and growth‑modification therapy.
While finite element analyses (FEA) are frequently employed to demonstrate that certain devices theoretically generate larger stress and deformation patterns—suggesting potentially more favorable treatment outcomes—such findings require careful interpretation. FEM studies have proven valuable for understanding force distribution and optimizing appliance design, particularly in comparing bone‑anchor effectiveness, and FEM has been validated for orthodontic tooth movement. However, Class III protraction therapy involves complex biological and developmental processes that extend beyond biomechanics alone.
Therefore, although favorable FEM results are informative, they cannot independently predict clinical success. Our study is the first to investigate whether larger deformation patterns, as predicted by FEM, effectively translate into improved treatment outcomes, and it is also the first to incorporate patient‑specific FEM in combination with real clinical effects. These limitations and their implications have been thoroughly addressed in the Introduction, Discussion, and Conclusion sections of the manuscript.
Comment 6: Figure 7 and 8 - Images of stress distribution must be explained.
Response 6: We thank the reviewer for this helpful suggestion. Figure 7 presents the stress distribution, while Figure 8 illustrates the deformation patterns along the three coordinate axes. To improve clarity, the captions of both figures have been revised to enhance readability and facilitate a better understanding of the results.
Comment 7: How could future predictive models should incorporate a combination of biological markers, growth patterns, and patient-specific tissue responses to more accurately reflect the complex nature of craniofacial adaptation?
Response 7: This sentence was removed from the revised manuscript, as it was deemed redundant and did not contribute additional information or substantive content. It was replaced by a section on the clinical relevance of the study.
Reviewer 3 Report
Comments and Suggestions for Authors
The aim of the present reaserch was to compare the finite element model predictions and actual clinical outcomes in class III protraction therapy.
Why the number of patients was so little?
Regardint this affirmations -The differences between the deformed and baseline models were analyzed using part 230
comparison analysis (PCA).- could the authors please specify what were the main differences encountered?
Please specify what were the limitations of the present research?
Originality: The article contains new and important information adequate to justify its publication.
Fit to the scientifical literature: The present paper demonstrates an adequate understanding of the relevant literature.
Methodology: The paper's argument was built on an appropriate base of theory, concept. The methods employed are appropriate and the statistics is well designed.Results:The results are presented clearly, concise, and precise.Discussions:The results analysed appropriately and the conclusion is adequately tie together the other elements of the paper. I would suggest the conclusion to be just one paragraph.Please also include more recent published articles.Implications for research, practice and/or society The paper clearly identify the implications for research, practice and also bridge the gap between theory and practice. These implications consistent with the findings and conclusion of the paper.Quality of Communication: The paper clearly present its case, in an appropriate technical language of the field and at the expected knowledge of the journal's readership.
The article is very interesting well written and respects all the norms of writing a scientific article.
Comments on the Quality of English Language
Moderate
Author Response
Comment 1: Why the number of patients was so little?
Response 1: The image data employed in this analysis were originally obtained within the framework of a randomized controlled trial (RCT) with extended long-term follow-up (Meyns et al, Eur J Orthod 2025 Feb / Meyns et al. Prog Orthod. 2025 Apr). Computed tomography (CT) scans were acquired at different time points. As this investigation was designed as a proof‑of‑concept study, the primary objective was not the immediate validation of the finite element model (FEM). Such validation is inherently time‑consuming, and at the outset it was uncertain whether the model would adequately reflect the actual treatment effect. Consequently, the study was initiated with four patients and subsequently expanded to include an additional six, resulting in a total of ten participants. The authors acknowledge that this represents a limited sample size; however, this was a deliberate choice consistent with the exploratory nature of a proof‑of‑concept study. This limitation, and its implications, are explicitly addressed in the Limitations section (line 611 – 614)
Although this proof-of-concept study involved a small sample size, the consistent discrepancy between model predictions and clinical results underscores the reliability of the findings. Future research should expand to include a larger patient cohort and consider additional factors beyond anatomical properties.
Comment 2: Regarding this affirmations -The differences between the deformed and baseline models were analyzed using part comparison analysis (PCA).- could the authors please specify what were the main differences encountered?
Response 2: We sincerely thank the reviewer for their thoughtful comments and the opportunity to clarify our approach. In this proof‑of‑concept study, we employed both qualitative and quantitative analyses to compare modeled deformation (PCAmD) with actual one‑year deformation (PCAaD). The qualitative analysis utilized color‑coded deformation maps, which, although not readily quantifiable, provided valuable visual insights into deformation patterns. To complement this, we performed a quantitative analysis based on five anatomical landmarks per patient in the upper jaw, thereby enabling a more objective evaluation relevant to protraction therapy (anterior displacement in the upper jaw region). The corresponding results are presented in Table 1, where we added mean and SD. We added the complete data set to a supplementary file 2.
The overarching aim of this work was to explore whether modeled deformation corresponded to actual deformation, thereby providing an initial step toward FEM validation. Both Pearson and Spearman correlation tests have been incorporated into the revised manuscript to strengthen the analysis (line 485 – 488):
Statistical testing revealed a significant negative correlation between PCAmD and PCAaD (Pearson’s r = –0.517, p = 0.003; Spearman’s ρ = –0.406, p = 0.026), indicating that higher modeled values were paradoxically associated with lower observed deformations.
Comment 3: Please specify what were the limitations of the present research?
Response 3: The limitations are further elaborated on in the discussion section (line 582 – 599):
This discrepancy highlights significant model limitations: 1. Oversimplified anatomy (omitting sutures, teeth, and periodontal ligaments) 2. Separate analysis of upper and lower jaws, ignoring their interconnected nature and the documented remodeling of the mandibular condyle and glenoid fossa during treatment [53] 3. Assumption of homogeneous bone properties 4. Exclusion of biological processes (growth, bone remodeling and post-treatment relapse) 5. Omission of muscular, masticatory, and soft tissue effects. These limitations suggest the need for more sophisticated models incorporating patient-specific data and biological processes. Another limitation of this study was the absence of mesh convergence testing, which is essential to validate the numerical stability and accuracy of finite element models. The selected mesh sizes were based on prior studies and anatomical considerations; however, future research should include systematic convergence testing to ensure that the results are independent of mesh density. Such testing will enhance the reliability of finite element models for clinical applications. While applying forces directly to nodes simplifies the modeling process, the authors recognize that selecting a surface, coupling it to a reference point, and applying the force to the reference point is an alternative approach that can more accurately distribute forces over a region. Future studies may explore this method to further refine loading conditions and improve the biomechanical representation of force applications in FEMs.
Reviewer 4 Report
Comments and Suggestions for Authors
Dear Authors,
This my opinion after I read your paper:
Abstract: Is not very clear the age of patients. At the end you mentioned “groing Class III patients” but the age of patients taken in study is not clear.
Introduction: References are relevant showing that authors have read the literature. Please use shorter sentences for a better understanding of the text and to be easy to follow. Even with this small observations Introduction is showing the relationship between biomechanics and biology, strengthning the multidisciplinary character of the study.
Material and Methods: Is respecting methodological rigor. Again sentences are to long and a little difficult to follow.
Results: Although interindividual variation are mentioned, they are not statistically quantified . Please add tables with standard deviations on the full sample.
Is not presented a statistical test for the differences in stress / strain between treatments.
Why didn’t you apply correlation test(Spearman/Person) between PCAmD and PCAaD? Or I did not find them in the article?
You have a small number of patients and this is attracts low statistical power.
Clinical significance is moderate, FEM useful for understanding forces not for clinical prediction.
Discussion: Are correct written and logical. You need to make more comparisons with litarature- the use of FEM. Explicitly mentioning the studies that attempted clinical validation (even if few) would strengthen the authors' position that this is the “first direct validation study.”
References: Please use the same letter style .
Author Response
Comment 1: Abstract: Is not very clear the age of patients. At the end you mentioned “growing Class III patients” but the age of patients taken in study is not clear.
Response 1: The patients included in the study were between 7 and 11 years of age, comprising five males and five females. This information was originally reported in the Materials and Methods section and has now also been incorporated into the abstract for completeness.
Comment 2: Introduction: References are relevant showing that authors have read the literature. Please use shorter sentences for a better understanding of the text and to be easy to follow. Even with this small observations Introduction is showing the relationship between biomechanics and biology, strengthening the multidisciplinary character of the study.
Response 2: The Introduction section has been revised to enhance clarity and improve readability:
Growing skeletal Class III malocclusion presents a complex challenge in orthodontics. The main difficulty lies in predicting growth potential of the maxilla and mandible. Early intervention decisions are complicated by uncertain skeletal growth patterns. Current debates focus on which patients are best suited for early Class III treatment [1–4].
Facemask (FM) therapy combined with rapid palatal expansion (RPE) is a common treatment approach [5]. The effectiveness depends on how forces transfer to the jaw. Traditional tooth-borne (TB) applications produce limited skeletal effects. Skeletal anchorage (SA) devices offer an alternative. These include mentoplate and palatal screws, such as Hybrid Hyrax [6]. SA devices may provide better skeletal results and vertical control, especially in high-angle patients [7–10].
Despite increasing use of SA devices, several questions remain unanswered. The optimal treatment age is unclear. Force levels are not standardized. Evidence comparing outcomes between SA devices and conventional methods is limited [11–14].
Initial treatment success doesn't guarantee long-term stability. Some patients show deterioration in occlusion and facial aesthetics as they mature [4]. The key challenge is patient selection. Clinicians need to identify which patients will maintain positive outcomes from early intervention. They also need to determine which patients should wait for skeletal maturity and possible orthognathic surgery.
Predictive models exist using 2D cephalometric and CBCT measurements. However, most are based on retrospective studies without proper validation [15–17]. New approaches combine these measurements with initial treatment response [18]. The predictive accuracy of these methods remains uncertain [15].
Research on maxillary protraction has evolved significantly. Early studies used basic models like wax, elastic, and dry skulls. Modern studies employ sophisticated 3D imaging and finite element (FE) modeling. Early research established fundamental concepts about centers of resistance and rotation [19–25]. Current FE analysis has enhanced our understanding of force distribution, particularly with SA techniques [26–29].
Despite the potential of finite element (FE) studies in analyzing skeletal anchorage (SA) techniques, several critical limitations persist in current research. Existing studies present contradictory findings regarding skeletal effects and vertical control [27,28], with many analyses overlooking crucial aspects such as mandibular deformation [26–28,30]. Research has predominantly concentrated on appliance design optimization while neglecting patient-specific anatomical variations. Furthermore, while studies frequently describe deformation patterns and hypothesize about their relationship to treatment outcomes, a significant knowledge gap remains: No studies have validated these models against actual clinical outcomes in Class III protraction therapy. Treatment success depends on both biomechanical forces and complex biological responses. Current modeling approaches may not capture all these factors. Understanding whether FEM can reliably predict treatment outcomes is crucial. This knowledge would either validate current biomechanical approaches or reveal the need for more sophisticated predictive tools.
This study presents the first direct comparison between finite element model predictions and actual clinical outcomes in Class III protraction therapy. We tested two key hypotheses using image data from a previous randomized controlled trial. First, we examined whether simplified FEM could accurately predict stress distribution and deformation patterns corresponding to actual treatment effects. Second, we investigated whether different treatment techniques (FM vs MP) would produce distinct patterns of skeletal deformation. Our findings challenge current assumptions about using biomechanical modeling in orthodontic treatment planning.
Comment 3: Material and Methods: Is respecting methodological rigor. Again, sentences are too long and a little difficult to follow.
Response 3: The Material and Methods section has been revised to enhance clarity and improve readability.
2.1Trial registration and ethical approval
We obtained CT scans from patients enrolled in a previous randomized controlled trial. This trial evaluated both 2D [31] and 3D [32] outcomes. The RCT was registered at www.ClinicalTrials.gov (ID: NCT02711111). The Ethics Committee at Ziekenhuis Oost Limburg, Belgium granted approval (EudraCT B371201629565) on December 13, 2016.
2.2Subjects
Our study analyzed CT scans from ten Caucasian patients with Class III skeletal malocclusion. The group included five females and five males, aged 7-11 years. All patients participated in a randomized trial comparing Facemask (FM) (Figure 1) and Mentoplate (MP) (Figure 2) treatments [31,32]. Each patient received a Hybrid Hyrax apparatus (HH) assembled with two mini-screws in the anterior palate and an expansion screw attached to first molar bands (Figure 1 and 2). We used the Alt-RAMEC protocol for expansion [33], wherein the HH was activated by the patient’s parents twice daily (0.25 mm per turn, two turns in the morning and two turns at night) for one week, followed by deactivation twice daily (two turns in the morning and two turns at night) for the next week. This cycle of alternating activation and deactivation was repeated three times. In the following week, the maxilla was adjusted to the suitable transverse dimension. The FM group received elastic forces of 360-400g per side and wore the appliance 12-14 hours daily. The MP group received 185g force per side with continuous wear.
We randomly selected two patients from each treatment group to develop our simplified Finite Element Model (FEM). We then tested the model on six additional patients, three from each treatment group, to evaluate its predictive abilities.
Our methodology followed three main phases. First, we created patient-specific 3D models from pre-treatment CT scans. Next, we performed finite element analysis to simulate both treatments. Finally, we compared predicted outcomes with actual one-year clinical results. This approach allowed direct validation of FEM predictions against real treatment outcomes.
2.3Creating a patient specific 3D FE model
We obtained CT scans at treatment (T0) start using a Siemens Somatom Force scanner (Siemens®, Erlangen, Germany). The scanning protocol used a slice thickness of 0.6 mm, with 50 mA current and 150 kV voltage. Each scan took 2.04 seconds to complete.
We imported the DICOM files into Mimics® software for segmentation. To improve efficiency, we excluded several regions not critical to the analysis. These included the posterior midface, skull, teeth, and superior portion of the alveolar process (Figure 3). The teeth were also removed since forces were transferred through a bone-anchored device (Hybrid Hyrax), which has been proven to produce good skeletal anchorage with minimal dento-alveolar effect [10,27]. The periodontal ligament was not simulated, based on a previous study suggesting that modeling the periodontal ligament in finite element analyses of skulls can be ignored if the values of stress and strain in the alveolar region are not required [34]. Finally, the superior portion of the alveolar process was excluded because significant remodeling occurs in this region during tooth eruption, which could interfere with accurate comparisons to the one-year follow-up scan.
We transferred the 3D model to 3-Matic software (Materialise®, Leuven, Belgium) for meshing. A surface mesh was generated using specific edge lengths tailored to the anatomical features of the structures: 1 mm for the maxilla and 2 mm for the mandible. These sizes were chosen based on previous studies [27] and to balance accuracy and computational efficiency, as the maxilla’s finer bone structures required higher resolution, while the mandible’s thicker and less complex geometry allowed for a coarser mesh without compromising accuracy. Different triangle edge lengths were used to maintain anatomical accuracy. The meshing process also involved ensuring that the surface mesh was free from elements with poor computational properties, such as sharp internal angles. Following this, a uniform volume mesh type was applied to both jaws for consistency during finite element analysis (FEA). The final 3D meshed FE models, depicted in Figure 3C, represent the maxillary and mandibular structures in detail, for further processing.
2.4Finite element analysis (FEA)
We conducted our analysis using Abaqus® software (Dassault Systèmes®, France). The model incorporated calculated force magnitudes and orientations along with appropriate boundary conditions. For this simplified FEM, the maxilla and mandible were modeled as uniform bone structures with isotropic and elastic material properties to ensure computational efficiency while maintaining physiological relevance. The assigned material properties were based on previous studies [26–29]. The maxilla was assigned a Young's modulus of 700 MPa and a Poisson's ratio of 0.3, representing its primarily trabecular bone composition, which is softer and more porous. In contrast, the mandible was assigned a Young's modulus of 1000 MPa and a Poisson's ratio of 0.3, reflecting its denser, cortical bone structure, which contributes to its increased stiffness.
2.5Boundary conditions
We applied boundary conditions for the upper jaw at the infra-orbital rim (Figure 4). This region remains stable during growth and treatment, making it an ideal reference point [35]. We fixed displacements in all three spatial directions at selected nodes while allowing rotation. The lower jaw presented more complex challenges for boundary conditions. Many regions undergo remodeling during growth. While the anterior chin and internal symphysis have been considered stable and reliable for voxel-based superimposition in growing patients [36], applying boundary conditions at these locations can overly constrain the model. Their high Young's modulus combined with boundary constraints prevented realistic deformation under treatment forces. The condylar region also proved unsuitable due to significant remodeling during growth and treatment [37]. We performed a three-dimensional volumetric comparison analysis on four patients from our RCT. This analysis revealed the angle of the mandible as a relatively stable area. We therefore fixed three nodes at each mandibular angle to achieve optimal model stability while allowing deformation. This approach balances the mandible's high stiffness (Young's Modulus) with realistic movement, as more constraints would prevent deformation and fewer would create instability (Figure 4).
2.6Magnitude and vector of the forces
The MP therapy used continuous low-force elastics generating approximately 200g of force per side [6,9,38]. We calculated precise force vector directions using 3-Matic software by analyzing the line between the MP's mandibular attachment point and molar anchor points. This patient-specific calculation ensures accurate modeling of force orientation in FE simulations. FM therapy required higher force elastics generating approximately 400g per side, worn 12-14 hours daily. We positioned the force vector at 25-30° to the occlusal plane [1,39,40]. Both treatments incorporated Alt-RAMEC using a HH expander with dual palatal mini-screws. This device generated approximately 2500g of force during palatal expansion [41–43].
2.7Force simulation
The forces acting on the upper and lower jaws were applied to the FE model through node sets to ensure accurate simulation of treatment forces (Figure 4). For the upper jaw, we selected two primary node sets based on the HH device design. The first set resided in the first molars' apical region. The second set included nodes near the anterior palatal screws. This configuration allowed for realistic simulation of both FM and MP therapy forces in the upper jaw.
In the lower jaw, the method of force application differed between FM and MP therapies. For FM therapy, 43 nodes were selected within the chin-cup contact area (Figure 4a). These nodes were distributed to cover the interaction zone where the FM applies pressure to the mandible, ensuring that the force transmission was realistically simulated. For MP therapy, each screw of the MP was represented by six nodes—three on the external surface and three internally within the mandible (Figure 4b). This arrangement was designed to accurately model the dispersive effect of forces as they propagate through the bone tissue surrounding the screws. The models incorporated realistic forces, with MP therapy requiring twice the elastic wear time of FM therapy. The number of deformation cycles was adjusted accordingly for each treatment to match actual clinical usage patterns. To capture these variations, the FM models were subjected to 5 deformation cycles, while the MP models underwent 10 cycles. Abaqus software enabled the iterative reapplication of forces to the already deformed geometry of the models, further enhancing the patient-specific accuracy of the FEA. This approach ensured that the simulation realistically represented the cumulative deformation and stress patterns corresponding to each treatment protocol.
The differences between the deformed and baseline models were analyzed using part comparison analysis (PCA). This technique produces color-coded maps that visualized the extent of deformation across various regions of the jaw structures. By incorporating these methods, the simulation captured the biomechanical behavior of craniofacial structures under treatment, providing insights into force distribution and the resulting deformations with high precision. Supplementary file 1 provides more details on node sets and forces applied on these nodes.
2.8Actual treatment effect
The baseline (T0) and one-year follow-up (T1) CT datasets were imported into Amira® software (version 2019.1, Thermo Fischer Scientific®, Merignac, France) in DICOM format to analyze skeletal changes over the treatment period. Using volume rendering, the datasets were visualized in 3D to enable precise identification of structural differences (Figure 5). A rigid voxel-based registration was performed with mutual information as the alignment metric, ensuring accurate superimposition of the datasets [44,45]. The T1 dataset was registered to T0 using stable anatomical landmarks: the inferior orbital rim for the maxilla and both the external oblique ridge and mandibular angle for the mandible. The inferior orbital rim's stability is well-documented in literature [35], while our previous PCA analysis confirmed the stability of the mandibular landmarks during growth. We then imported the registered datasets into Mimics® software (version 20.0, Materialise®, Leuven, Belgium) for segmentation. A semi-automatic thresholding approach, refined with manual adjustments, was employed to construct high-resolution 3D volumetric surface models of the maxilla and mandible. These models were then exported in STL format (Figure 5), preserving the geometric fidelity required for further analysis.
The STL models were subsequently transferred to 3-Matic® software (version 14.0, Materialise®, Leuven, Belgium) for part comparison analysis. This process calculated the mean differences between the T0 and T1 datasets, quantifying treatment-induced changes in the skeletal structures. The results were visualized as a color-coded map (Figure 5), with different colors representing the magnitude and distribution of deformation.
2.9Comparing modeled deformation with actual deformation
We evaluated FEM accuracy by comparing predicted (PCAmD) and actual (PCAaD) deformations using visual and quantitative methods. Color-coded maps illustrated deformation patterns. Initial analysis of four FEM cases showed promising maxillary results, but mandibular comparisons proved impossible due to model limitations. We expanded the study with six additional patients. We focused on maxillary deformation using five anatomical landmarks per patient for qualitative analysis (Figure 6, Table 1). Our analysis revealed that predicted deformation values were notably smaller than actual measurements. The correlation between PCAmD and PCAaD at these landmarks provided insight into model accuracy.
Comment 4: Results: Although interindividual variation are mentioned, they are not statistically quantified. Please add tables with standard deviations on the full sample.
Response 4: We sincerely thank the reviewer for their thoughtful comments and the opportunity to clarify our approach. In this proof‑of‑concept study, we applied both qualitative and quantitative analyses to compare modeled deformation (PCAmD) with actual one‑year deformation (PCAaD). The qualitative analysis employed color‑coded deformation maps, which, although not directly quantifiable, provided valuable visual insights into deformation patterns. To complement this, we conducted a quantitative analysis using five anatomical landmarks per patient in the upper jaw, thereby enabling a more objective assessment relevant to protraction therapy (anterior displacement in the maxillary region). The corresponding results are presented in Table 1, where we added mean and SD. We added the complete data set to a supplementary file 2.
The overarching aim of this work was to examine whether modeled deformation corresponded to actual deformation, thereby providing an initial step toward FEM validation. To strengthen the robustness of this analysis, both Pearson and Spearman correlation tests have been incorporated into the revised manuscript. We also acknowledge the limited sample size, which reflects the exploratory nature of a proof‑of‑concept design, and this limitation has been explicitly addressed in the revised version of the manuscript.
Comment 5: Is not presented a statistical test for the differences in stress / strain between treatments.
Why didn’t you apply correlation test (Spearman/Person) between PCAmD and PCAaD? Or I did not find them in the article?
Response 5: We would like to sincerely thank the reviewer for this valuable comment. In response, both Pearson and Spearman correlation tests have been incorporated into the revised manuscript to strengthen the analysis (line 485 – 488):
Statistical testing revealed a significant negative correlation between PCAmD and PCAaD (Pearson’s r = –0.517, p = 0.003; Spearman’s ρ = –0.406, p = 0.026), indicating that higher modeled values were paradoxically associated with lower observed deformations.
Comment 6: You have a small number of patients, and this is attracting low statistical power.
Response 6: The image data employed in this analysis were originally obtained within the framework of a randomized controlled trial (RCT) with extended long-term follow-up (Meyns et al, Eur J Orthod 2025 Feb / Meyns et al. Prog Orthod. 2025 Apr). Computed tomography (CT) scans were acquired at different time points. As this investigation was designed as a proof‑of‑concept study, the primary objective was not the immediate validation of the finite element model (FEM). Such validation is inherently time‑consuming, and at the outset it was uncertain whether the model would adequately reflect the actual treatment effect. Consequently, the study was initiated with four patients and subsequently expanded to include an additional six, resulting in a total of ten participants. The authors acknowledge that this represents a limited sample size; however, this was a deliberate choice consistent with the exploratory nature of a proof‑of‑concept study. This limitation, and its implications, are explicitly addressed in the Limitations section (line 611 – 614):
Although this proof-of-concept study involved a small sample size, the consistent discrepancy between model predictions and clinical results underscores the reliability of the findings. Future research should expand to include a larger patient cohort and consider additional factors beyond anatomical properties.
Comment 7: Clinical significance is moderate, FEM useful for understanding forces not for clinical prediction.
Response 7: We sincerely thank the reviewer for their thoughtful comments and the opportunity to clarify our approach. The Discussion section (lines 570–571) has been expanded with additional references to recent literature to highlight the novelty of this study as the first direct validation attempt in this field. It is indeed exceptional to have access to a prospectively collected three‑dimensional dataset of children during growth and growth‑modification therapy. While finite element analyses (FEA) are frequently employed to demonstrate that certain devices theoretically generate larger stress and deformation patterns—suggesting potentially more favorable treatment outcomes—such findings require careful interpretation. FEM studies have proven valuable for understanding force distribution and optimizing appliance design, particularly in comparing bone‑anchor effectiveness, and FEM has been validated for orthodontic tooth movement. However, Class III protraction therapy involves complex biological and developmental processes that extend beyond biomechanics alone. Therefore, although favorable FEM results are informative, they cannot independently predict clinical success. Our study is the first to investigate whether larger deformation patterns, as predicted by FEM, effectively translate into improved treatment outcomes, and it is also the first to incorporate patient‑specific FEM in combination with real clinical effects. These limitations and their implications have been thoroughly addressed in the Introduction, Discussion, and Conclusion sections of the manuscript.
Comment 8: Discussion: Are correct written and logical. You need to make more comparisons with literature- the use of FEM. Explicitly mentioning the studies that attempted clinical validation (even if few) would strengthen the authors' position that this is the “first direct validation study.”
Response 8: We thank the reviewer for raising this important point. To the best of our knowledge, no studies to date have undertaken a clinical validation of FEM in Class III protraction therapy. The limited number of existing studies in this field have primarily focused on appliance design and its influence on deformation patterns, as cited in the Discussion section. However, none of these investigations have attempted a direct clinical validation, nor have they employed patient‑specific FEM approaches.
Comment 9: References: Please use the same letter style .
Response 9: was adjusted in the revised manuscript.